# The Effect of W Content on the Microstructure, Mechanics and Electrical Performance of an FeCrCo Alloy

**DOI:** 10.3390/ma16124319

**Published:** 2023-06-11

**Authors:** Huiqi Wang, Hui Zhang, Mangxian Liu, Jianqun Liu, Zhipeng Yan, Changming Zhang, Yapeng Li, Junjun Feng

**Affiliations:** 1School of Materials Science and Engineering, Shaanxi University of Technology, Hanzhong 723000, China; whq1998113@163.com (H.W.); fengjunjun_shaanxi@163.com (J.F.); 2Avic Zhonghang Electronic Measuring Instruments Co., Ltd., Hanzhong 723000, China; mangxianl521@163.com (M.L.); jianqunl521@163.com (J.L.); zhipengy521@163.com (Z.Y.); 3School of Mechanical Engineering, Shaanxi University of Technology, Hanzhong 723000, China; zhangchangmingsx@126.com

**Keywords:** FeCrCoW alloy, microstructure, mechanical property, electrical performance, martensite

## Abstract

In this paper, FeCrCoW alloys with different W contents (0.4, 2.1 and 3.4 at%) are designed and studied in order to overcome the existing shortcomings of resistance materials. These resistance materials have high resistivity and a low temperature coefficient of resistivity. It is observed that the addition of W has a remarkable effect on the phase structure of the alloy. In particular, when the W content is 3.4 at%, the single BCC phase of the alloy can be transformed into the BCC and FCC phase. Meanwhile, when analyzed by transmission electron microscopy, there are stacking faults and martensite in FeCrCoW alloy with W content of 3.4 at%. These features are related to excessive W content. In addition, the strength of the alloy can be improved, and the ultimate tensile strength and yield strength are both very high, which are considered as grain-boundary strengthening and solid solution strengthening, caused by the addition of W. The electrical resistivity of the FeCrCoW alloys decreases when the content of W is more than 2.1 at%. The maximum resistivity of the alloy is 170 ± 1.5 μΩ·cm. Moreover, the unique properties of the transition metal allow the alloy to have a low temperature coefficient of resistivity in the temperature range of 298~393 K. The temperature coefficient of resistivity values of the W0.4, W2.1 and W3.4 alloys are −0.0073, −0.0052 and −0.0051 ppm/K. Therefore, this work provides a vision for resistance alloys, which can achieve highly stable resistivity and high strengths in a certain temperature range.

## 1. Introduction

Resistive materials are generally characterized by a high resistivity accuracy, which is greater than 100 μΩ·cm, and a low temperature coefficient of resistivity (TCR) [1]. Resistance alloys, as the core materials of resistive sensors, are applied to precision parts in aerospace, national defense, automotive electronics, mechanical equipment and industrial automatic control fields [2,3,4,5,6,7]; these fields are all in need of high-precision electronic measurement systems, such as the Global Positioning System, thermoelectric devices, and temperature and flow control sensors for automotive and aircraft applications. At present, the method used to create alloys is widely applied in order to improve the electrical resistivity of metal materials. It has been demonstrated that electron scattering can enhance electrical resistivity due to crystal lattice distortion [8]. For instance, by adjusting the proportions of three elements added to a FeCrAl alloy, the alloy’s resistivity and TCR values were significantly improved. Defects and solute elements tend to diffuse around grain boundaries, resulting in an increase in resistivity [9]. They therefore serve as resistance materials because of their excellent electrical resistivity and cost efficiency [10].

Generally, resistance alloys are composed of transition metals. Such as Cu–Ni, and Cu–Ni–Mn alloys currently widely serve as resistive materials [11]. In addition, high-entropy alloys such as FeCoNiPd, FeCoNiPt, FeCoNiTaAl and Al_2.08_CoCrFeNi HEAs, as a class of alloy with multiple main elements, can cause obvious lattice distortion and promote the scattering of conduction electrons [12,13,14]. This is also a hopeful candidate material for realizing high resistivity. However, the accuracy and potential contexts of their applications are limited, as it is difficult for the resistivity to meet the requirements of special physical environments. It is well known that FeCrCo alloys have outstanding plasticity and machinability. They can be machined into small equipment that is suitable for generating a precise sizes and complicated shapes, such as wires, tubes, bars and strips. Accordingly, they are widely employed in other fields, such as electrical appliances and hybrid cars [15,16,17]. At present, the magnetic performance of FeCrCo alloys has been discussed in detail. However, in other aspects, there are not enough published documents referring to the electrical performance of FeCrCo alloys. Yet, as industry develops, electronic devices are being miniaturized, and resistance materials with excellent electrical properties are thus required to boost machinability [18]. It has been realized that lattice defects can greatly affect and promote resistivity to a certain extent [19]. Aisaka Tsuyoshi et al. reported [20] that W, as a transition metal, has a positive influence on electrical resistance. The atomic radius of W is much larger than that of the base alloy. When W is dissolved into an FeCrCo alloy, it causes more lattice distortion and enhances electron scattering. Therefore, in this paper, we attempt to develop a resistance material with high resistivity and a low-resistance temperature coefficient by adding W to an FeCrCo alloy. The electrical and mechanical performances of the FeCrCoW alloys are expected to be closely associated with their microstructure. However, there are few reports on the interaction between the microstructure and mechanical and electrical properties.

Taking the above into account, this work discusses the mechanical performance and mechanisms involved in the improvement of the electrical resistivity of the FeCrCoW alloys based on the experimental results. We focus on studying the changes in the resistivity of the alloy. From room temperature to 393 K, the TCR of the alloys is compared to other bulk alloys. Meanwhile, the reasons for the high resistance of the alloys and the variations in the TCR with temperature are analyzed deeply. This study is not only conducive to a greater overall awareness of the basic properties of alloys, but also has important implications for enlarging the application of alloys. The research provides beneficial ideas for the development of precision resistance alloys with high-cost effectiveness, high stable resistivity, and a high strength within a certain temperature range.

## 2. Experiments

### 2.1. Material Preparation

In this experiment, the FeCrCoW (x = 0.4, 2.1 and 3.4 at%) alloys were denoted as W0.4, W2.1, and W3.4, respectively, and their chemical compositions are represented in Table 1. All alloys were prepared via vacuum arc smelting and used small pure metals with a purity exceeding 99.8 wt.%. During the process, a furnace was filled with high-purity argon, which was used as a protective gas. The smelting was carried out at 1500 °C and cooled with the furnace. This process was repeated six times to ensure that the components and elements in the alloy were homogeneous. The liquid metal was cast in a metal mold at 1600 °C with a 10 × 5 × 3 cm^3^ cube alloy ingot, as shown in Figure 1. The ingots were then homogenized at 900 °C and kept for 1 h [21]; Ar atmosphere served as a protective gas. A cube with a size of 1 × 1 × 1 cm^3^ was cut from the middle of the alloy in order to observe the microstructure and conduct performance tests.

### 2.2. Characterization

The phase constitution of the alloy was identified using X-ray diffraction (XRD, DX-2000, Shanghai Precision Instrument Co., LTD, Shanghai, China) tests with Cu *K_α_*_1_ radiation (*λ* = 0.15418 nm). The scanning area was between 40° to 90° (2*θ*) and the step size was 5°/min. Microstructural examinations of the alloy were carried out using optical microscopy (OM, IE500M, Ningbo Shunyu Co., Ltd., Ningbo, China) and grain size testing was performed using Nano Measurer 1.2. Prior to observation, all samples were metallographically polished and etched with the mixed solution; this was so that the volume ratio of HNO_3_ and HCl was kept constant at 1:3. The Vickers hardness (FM-700, FUTURE-TECH, Tokyo, Japan) of the alloy was measured under a load of 200 g for 10 s. Each sample was tested in seven positions and the average value was calculated. Meanwhile, the microstructures, composition and fracture morphology of the alloys were examined using a scanning electron microscope (SEM, FEI Quanta FEG250, Thermo Fisher Scientific, Waltham, MA, USA) equipped with energy-dispersive spectroscopy (EDS) instruments. In transmission electron microscopy (TEM, Talos F200X, Thermo Fisher Scientific, Waltham, MA, USA) tests, the sample was mechanically ground to 50 μm. Subsequently, the samples were further subjected to the electrolytic double-jet technique in a solution of 20% HClO_4_ and 80% C_2_H_5_OH at a temperature of −30 °C for 170 s. For electron backscattering diffraction (EBSD, SIGMA500, Zeiss, Oberkochen, Germany) measurements, all samples were prepared using electrolytic polishing. The EBSD images were collected at 20 kV. The working distance was 16 mm and the step size was 0.15µm. Before conducting a room-temperature tensile test, the samples were firstly cut into dog-bone-shaped pieces, and the gauge length, width, and thickness were 47 mm, 4 mm, and 2 mm, respectively. The experiment was conducted using a miniature control electronic universal testing machine (CMT-5015, Shanghai Qingji Instrument Technology Co., Ltd, Shanghai, China) with a strain rate of 1 × 10^−3^ s^−1^. In order to prevent the influence of local stress concentrations on the tensile properties of the sample, the tensile sample was polished with sandpaper before the test. The tensile test was repeated three times. Moreover, the resistivity of the alloy was characterized using a four-probe resistivity tester (HPS2662, Changzhou Helpass Electronic Technology Co., Ltd., Changzhou, China) at room temperature. The probe spacing was 1 mm. The sample was measured three times and the calculated average value was used as the resistivity of the alloy. The TCR of cuboid samples was measured by thermistor tester (RMS-1000PTC, Wuhan Huagong Xingaoli Electron Co., Ltd., Wuhan, China) under the condition of continuous heating from room temperature to 393 K. The size of the sample was 12 × 10 × 2 mm. During this process, a constant current of 10 mA was allowed to pass through the sample. The measurement for the precision of resistivity was 0.1 μΩ, and the heating rate was at 2 K/min. The experiment was repeated three times.

## 3. Results and Discussions

### 3.1. Microstructure Analysis

Figure 2 represents the XRD patterns of the FeCrCoW alloys with different contents of W. The results suggest that W0.4 and W2.1 alloys are consisted of a BCC-*α* phase. The main crystal plane of (110), (200) and (211) were present. With the content of W element reaching 3.4 at%, the intensity of the BCC-*α* phase gradually decreased, while the FCC-*γ* phase appeared and increased. The main crystal plane of (110), (200), (211), (111), (200) and (220) were present. There was no precipitated phase in the XRD plot. In particular, the diffraction peak of the alloy with a W content of 0.4 at% was narrow and high, which indicates that the grain size was coarse and the defect concentration was low. With the increase in the W content, the full width at half maximum (FWHM) became wider, indicating that the lattice distortion increased. No diffraction peaks were detected for the alloys containing W can be found in Figure 2. This phenomenon means that W element already exists in the solid solution as a substitution element, causing the distortion of the matrix unit cells [22]. The lattice parameter of the strongest peaks of the phases in the XRD patterns can be calculated as follows [23]:(1)$$d=\frac{a}{\sqrt{h^{2}+k^{2}+l^{2}}}$$
where *h*, *k* and *l* represent the Miller indices, *d* represents interplanar spacing, and *a* represents the lattice constant of the unit cell. It is worth noting that the lattice parameters are much higher than the standard lattice parameters with the change in the content of W. From the position of the peak, it can be inferred that the lattice parameters are 0.2870 nm, 0.2878 nm and 0.2877 nm, respectively (Table 2). Compared to the standard lattice parameters at 0.2866 nm [24], the formation of the solid solution of FeCrCoW may be emphasized. The bonding force between is determined by the melting point of the alloy. Generally, alloys with high melting points enhance the interaction forces between atomic bonds [25]. W atoms dissolve into the matrix of FeCrCoW, resulting in more severe lattice mismatch. Hence, the lattice parameter varies along with the content of W.

Figure 3 shows the OM microstructure of the FeCrCoW alloys. The surfaces of the samples show different contrast degrees under the metallographic microscope, which is mainly due to the different grain orientations leading to differences in the corrosion degree. When the W content was 0.4 at%, the average grain size was 439.22μm. The flake of the BCC phase with a grey part was obviously observed after corrosion The average grain size decreased to 390.28 μm with the increase in the content of W to 2.1 at%. The sub-grains within the grains are striped. With the content of W further increasing to 3.4 at%, the average grain size decreased to 363.27 μm. The internal grains possessed slatted martensite (indicated by red arrow), as displayed in Figure 3c. No diffraction peak for martensite was found in the XRD analysis, which may be due to the low content of martensite. The TEM characterization of martensite will be further discussed. Above all, it can be observed that the grain size gradually decreased as the content of W increased. This result indicates that the increase in W content plays a significant role in grain refinement.

The SEM images and EDS mapping of all alloys are shown in Figure 4. It is evident that W was dissolved into the FeCrCo alloy. At the same time, the distribution of each element was uniform.

As displayed in Figure 5, the W2.1 and W3.4 alloys were further characterized via EBSD analysis. Compared to the W2.1 alloy with a BCC-*α* phase, the W3.4 alloy is composed of a BCC phase and an amount of FCC-*γ* phase. The corresponding proportions are 54.1% and 39.9% in Figure 5b, respectively. Figure 5(a1,b1) shows the inverse pole figure (IPF) maps. As can be seen from Figure 5(b1), there are plate-like structures similar to twins in the FCC phase (corresponding to the right side, black arrow). Figure 5(b4) shows the local EBSD diagram of the W3.4 alloy. The yellow part and the blue part are twinned to each other. The twins are lamellar and the twin boundary is relatively straight. The twin boundary difference is 60° and the twin plane is $\left(\overset{-}{1}11\right)$, as shown in Figure 5(b5). We speculate that the appearance of twins is closely related to the decrease in the layer fault energy caused by the addition of W. A twin boundary can not only inhibit dislocation motion and increase strength, but can also be used to coordinate the plastic deformation of the slip surface of dislocation, which is significantly important to improving the alloy’s strength and toughness [26]. Figure 5(a2,b2) shows the grain boundaries and phase boundaries. High-angle grain boundaries mainly exist in the BCC phase of the W3.4 alloy. Figure 5(a3,b3) shows the Kernel average misorientation (KAM) maps. The average KAM value of the W2.1 alloy is 0.21°. The FCC phase is blue with a low KAM value, while the BCC phase is green with a high KAM value, as shown in Figure 5(b3). The BCC phase has a higher mean orientation difference than the FCC phase in the W3.4 alloy, indicating that the BCC phase has a higher dislocation density [27]. The average KAM value of the W3.4 alloy is 0.16°.

TEM analysis characterized the structure of the phase in the FeCrCoW alloy. TEM images of the W2.1 alloy are shown in Figure 6a–b; these were obtained in order to reveal the structure of the enriched Cr particle in the alloy. Figure 6a shows the high-angle annular dark-field (HAADF) STEM image of the sample. The corresponding STEM–EDS maps (Figure 6(a1–a4)) for enriched Cr particles can be observed, and the average diameter of the enriched Cr particle is 23.62 nm. The fast flourier transformation (FFT) image (Figure 6b) obtained from the BCC matrix and the enriched Cr particle further identify the structures of the two phases. Spacing planes of 0.212 nm and 0.206 nm were measured by calibrating the FFT images, respectively. The mismatch between the enriched Cr particle and matrix can be calculated using the following formula [28]:(2)$$\delta =\frac{2\times (d_{matrix}-d_{particle})}{d_{matrix}+d_{particle}}\times 100\%=\frac{2\times (0.212-0.206)}{0.212+0.206}\times 100\%=2.9\%<5\%$$
where *d_matrix_* and *d_particle_* are the crystal plane spacing. The mismatch is less than 5%. Therefore, the particle is completely coherent with the matrix. The enriched Cr particle phase has an effect on the alloy’s electrical properties, which will be discussed below. Cross-stacking faults were obviously found in the W3.4 alloy, as shown in Figure 6c. Stacking faults (SF) are formed during lattice reconstruction for various reasons. The layer faults are extremely fine, and the spacing between them is only a few nanometers. Adding W can reduce stacking fault energy, which causes the lattice to easily slip along the close-packed planes [29]. This slip can lead to the occurrence of some dislocations and stacking faults, which can further evolve into twin boundaries [30]. This could shed more light on the twin formation, as shown in Figure 5(b4). In addition, reducing the layer fault energy is beneficial to the formation of martensite [31]. The stacking faults on both sides are composed of different phases. The selection diffraction in Figure 6(c1,c2) corresponds to the FCC phase and the body-centered tetragonal (BCT) martensite phase, respectively. We speculated that they are distributed along the C-axis, causing the superlattice (001) crystal face to appear in the middle of the (002) crystal face, as displayed in Figure 6(c2). It is obvious that the martensitic phase is formed by superimposed stacking faults. Figure 6d,e shows the slatted martensite. The formation of martensite is mainly related to the increase in the mobility of the grain boundary and the driving force of grain boundary migration caused by the addition of W [32].

### 3.2. Analysis of Mechanical Properties

The engineering stress–strain curves of the FeCrCoW alloy with different contents of W at room temperature can be seen in Figure 7a. The ultimate tensile strength (UTS), 0.2% offset yield strength (YS) and uniform elongation (UE) have been drawn from the engineering stress–strain curves and are shown in Table 3. As the content of W increases, it shows an upward trend. The YS and UTS of the alloy are 237 ± 2.7 MPa and 687 ± 4.9 MPa, with the addition of 0.4 at% W. With the content of W rising to 2.1 at%, the YS and UTS of the alloy increase to 301 ± 3.9 MPa and 793 ± 3.1 MPa. With the content of W rising to 3.4 at%, the YS and UTS are 320 ± 4.1 MPa and 956 ± 3.3 MPa. Adding W is good for improving the yield strength of the FeCrCoW alloy. Similarly, the fracture elongation of the FeCrCoW alloy is obviously affected by the content of W, and gradually increased with the increase in W. The fracture elongation values are 5.0 ± 1.2%, 6.3 ± 2.9% and 24.2 ± 2.0%, respectively. Figure 7b shows the changes in the Vickers hardness of the FeCrCoW alloy; the Vickers hardness values are 257 ± 1.6 HV, 271 ± 2.3 HV and 277 ± 0.9 HV, respectively. The presence of twins in the W3.4 alloy is conducive to improving its mechanical properties [33].

Figure 8 schematically illustrates the fracture morphologies of the alloys with different contents of W. With the content of W increasing, the number of dimples present shows a rising trend. There were only a few dimples in the alloy and the depth was shallow when the content of W was 0.4 at%. The deformation was mainly concentrated in the shear stress area, and the pores formed first elongated towards the shear stress area, showing a parabolic shape on the fracture surface. It was found that the fracture mode was plastic fracture. Small dimples and large dimples were distributed in the alloy when the content of W reached 2.1 at%, which corresponded to plastic fracture. In particular, the dimples were evenly distributed and became deep when the content of W increased to 3.4 at%, as shown in Figure 8c. The alloy slipped along the dimple wall under the combined action of tensile and shear stress [34], and showed higher ductile fracture. The results for fracture morphology are consistent with the observed results for fracture elongation. The W3.4 alloy was composed of BCC and FCC phases, and its high strength and plasticity were not only related to lattice distortion, but also attributed to the alloy with a mixture of BCC and FCC phases. In addition, the FCC phase in the alloy is beneficial for plasticity [35]. It is not difficult to find that the W3.4 alloy can achieve a balance of strength and plasticity by combining the advantages of the FCC phase and BCC phase.

In the current FeCrCoW alloy, a variety of microstructure characteristics, including solid solution matrix and grain boundaries, all are key factors for the strength. This study overlooked the impact of precipitation strengthening. In addition, dislocation strengthening also can be ignored, as none of the alloys experienced any plastic deformation before testing. It is well known that solid solution strengthening is closely related to lattice mismatches or lattice malformations in the solid solution structure [36]. This lattice distortion usually enhances lattice friction and thus increases the yield strength of the alloy [37]. Therefore, the yield strength varies with increasing W concentrations. For alloys added under different conditions, the contribution of solid solution strengthening is different. In this study, with the increase in the content of W, the yield strength of the W0.4 alloy increased by ~26% compared to that of the W3.4 alloy.

In addition, as the W content increases, the grain sizes gradually decreased, as can be seen from Figure 3. The grain sizes of the W0.4, W2.1 and W3.4 alloys were 439.22 μm, 390.28 μm and 363.27 μm, respectively. The yield strength increased with the increase in the content of W. As a result, grain boundary strengthening is also essential for the FeCrCoW alloy. The results show that the addition of W not only improves the lattice friction stress by improving the local lattice distortion, but also promotes the strengthening effect of grain refinement.

### 3.3. Analysis of Electrical Performance

#### Electrical Resistivity

Electrical resistivity, as an electrical parameter, is usually related to the chemical composition, alloy structure and working temperature of the material [38]. We discussed deeply and elaborated on the influence of the composition of alloys on their high resistivity at room temperature in this study of FeCrCoWx (x = 0.4, 2.1 and 3.4 at%) alloys. With the rising content of W, the electrical resistivity of the FeCrCoW alloy increased first and then decreased, as shown in Figure 9. The electrical resistivities were 160 ± 1.5 μΩ·cm, 170 ± 1.5 μΩ·cm and 141 ± 1.2 μΩ·cm, respectively. The resistivity of the alloy is related to the degree of defect, that is, the residual resistance in Matthiessen’s rule is affected [39]. Previous research has revealed that electrical resistivity is closely related to four scattering mechanisms, which include phonon scattering, interface scattering, dislocation scattering and impurity scattering [40,41]. In this study, there are two principal reasons for the high resistivity of the FeCrCoW alloy. On the one hand, the addition of W caused severe lattice distortion, leading to high electron scattering and enhanced resistivity. This is mainly because the atomic radius (0.137 nm) of W is much larger than the atomic radii of the other three elements; indeed, the atomic radii of Fe, Co and Cr are 0.126 nm, 0.125 nm and 0.128 nm, respectively [25]. W has a large atomic radius and slowly diffuses in the FeCrCo matrix [42]. Usually, W atoms randomly occupy lattice positions, resulting in an asymmetric surrounding environment for each atom, as well as differences in the size and bond energy between adjacent atoms, leading to atoms deviating from their lattice positions at different positions. Therefore, the strong scattering of electrons owing to lattice distortion inevitably leads to high resistivity [43]. It can be roughly said that adding W causes lattice distortion and leads to changes in resistivity. On the other hand, Fe, Cr, Co, and W are all transition metal elements that exhibit unique s-d scattering effects. The main characteristic of transition metals is that they have unfilled d bands, while conductive electrons include unfilled d bands and filled s bands. Conductive s electrons are scattered and transferred to the d band, which reduces the average free path of electrons and leads to an increase in the resistance of the FeCrCo alloy [44]. Therefore, compared to other metals, this s-d scattering behavior leads to the relatively high resistivity of transition metals.

In particular, the size of the enriched Cr particle was found in the W2.1 alloy. By increasing the volume fraction and size of nanoparticles, the surface area of conductive electrons increased, further promoting scattering and enhancing resistivity [18]. Previous research has shown that the orientation angles of grains have an influence on resistivity. When the orientation angle is smaller, the scattering effect is weaker [45]. The average KAM values of the W2.1 and W3.4 alloys are 0.21° and 0.16°, respectively. Therefore, compared with the W2.1 alloy, the resistivity of the W3.4 alloy decreased by 17%. In addition, for bulk metal materials, the scattering capacity of the twin boundary for free electrons is much lower than that of the ordinary boundary [46]. This is also one of the reasons why the resistivity of the alloy decreases. According to the obtained results, adding different contents of W to the FeCrCoW alloy can change the grain size of the alloy, introduce defects and further affect the resistivity. Obviously, there are multiple factors that can cause changes in the resistivity of alloys, some of which can enhance the resistivity, and others that can diminish it. As a result, the electrical resistivity always fluctuates.

The relation between resistivity–temperature in FeCrCo alloy with different W content was presented in Figure 10a; this was calculated using the formula listed below [44]:(3)$$TCR=\frac{{(\rho }_{T}-\rho_{R})}{\rho_{R}(T-T_{R})}$$
where *ρ_R_* represents the resistivity at room temperature, and *ρ_T_* represents the resistivity at another temperature (393 K). The resistivity for simple metals generated by electron phonon interactions typically shows a positive TCR value [47]. Yet, the present FeCrCoW alloy displays a negative TCR value within the temperature range of room temperature to 393 K. The TCR values of FeCrCo alloys with different W contents all fluctuated near zero. They had a small negative TCR range between −0.0073, −0.0052 and −0.0051 ppm/K. In this study, an electrical property (resistivity and TCR at room temperature) plot was constructed for the FeCrCoWx (x = 0.4, 2.1 and 3.4 at%) alloys and is shown in Figure 10b. In addition, the electrical properties of some commonly used alloys, such asNi_80_Cr_20_ [48] and FeCrAl_10_ [48], were also displayed, as well as the electrical properties of recently reported HEAs such as Al_0.28_CoCrFeNiCu [13], AlxCoCrFeNi [49] and Fe_55_Cr_28_Al_17_ [18], which were also shown for comparison. It can be clearly seen that the electrical performance of the FeCrCoW alloy in this study is superior to previous resistance alloys, suggesting that it has good application prospects as a high-accuracy resistance material.

For crystalline alloys, the electrical resistivity (*ρ*) can be shown by the below equation [44]:(4)$$\rho =\left(1-\gamma \right)\rho_{ip}+\rho_{0}e^{-2W^{*}}$$
(5)$$\gamma =\frac{2\pi }{\Lambda q_{D}}$$
where *ρ_ip_* represents the ideal one-phonon resistivity, *ρ*_0_*e*^−2*W**^ represents the elastic component, *ρ*_0_ represents residual resistivity, and 2*W*^*^ represents an averaged Debye–Waller exponent. When multi-phonon terms were neglected, which should drop off faster than *ρ*_ip_, Λ represents the electron mean free path and *q_D_* represents the Debye wave vector in Equation (4). From the formula, it can be seen that if the resistivity is high, Λ will decrease and produce a large *γ*, suggesting that the TCR of the phonon-scattering part will reduce. This also means that when the resistivity is large enough, small or negative TCRs may occur. In contrast, phonon scattering no longer changes with temperature when the initial resistivity reaches its limit. The Debye–Waller exponent will be affected. When the temperature increases, the residual resistivity ρ_0_ becomes smaller. The correlation between high resistivity and a low TCR are consistent with the saturation effect in electron transport [50]. The FeCrCoW alloys have high room-temperature resistivity. As a result, the alloy’s TCR values are no longer large positive values, but fluctuate near zero. In fact, materials with TCR ≈ 0 not only reduce the cost of new equipment, but also obviously raise sensitivity in a lot of prominent high-accuracy applications [10].

## 4. Conclusions

In this study, a resistance alloy with excellent comprehensive performance was obtained by adding W into the FeCrCo matrix. At the same time, the following conclusions can be drawn:The content of W is a key factor affecting the composition of FeCrCoW alloy phases. The FCC phase appears when the content of W increases. In addition, the layer fault energy of the alloy decreases with the addition of 3.4 at% W content, which leads to the occurrence of twins. Considering the improvement in strength and ductility, it was clearly indicated that W has a beneficial impact on a combination of properties in the FeCrCo alloy. W element plays a role in refining grain and strengthening the solid solution.The variation in the content of W was an important factor affecting the electrical resistivity of the FeCrCoW alloy. W exists in the alloy in the form of a solid solution atom. The maximum resistivity value is 170 ± 1.5 μΩ·cm with the addition of 2.1 at% W. The increase in the electrical resistivity caused by the solid solution was mainly due to the dissimilar atom’s dissolution, which caused the distortion of the solvent lattice.The change in the Kernel average misorientation and twin were also important factors affecting the resistivity of the alloy. The average KAM value of the W2.1 alloy is smaller than that of the W3.4 alloy. Therefore, compared with the W2.1 alloy, the resistivity of the W3.4 alloy decreased by 17%. Therefore, FeCrCoW alloys can be used as a promising resistive material.

## Figures and Tables

**Figure 1 materials-16-04319-f001:**
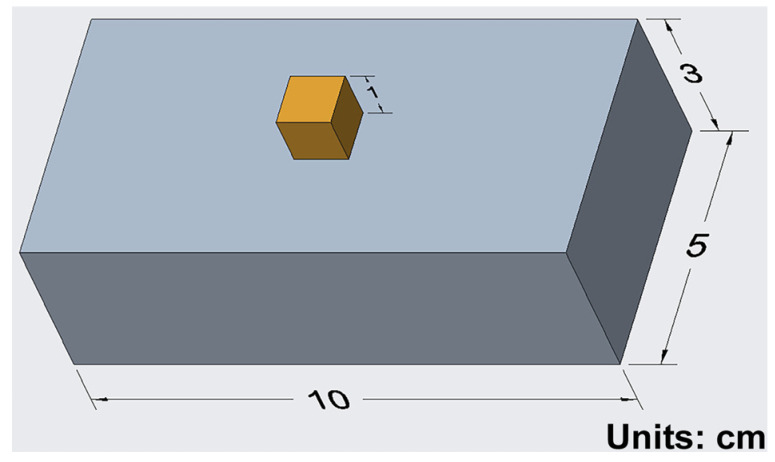
The macroscopic morphology of the FeCrCoW alloy.

**Figure 2 materials-16-04319-f002:**
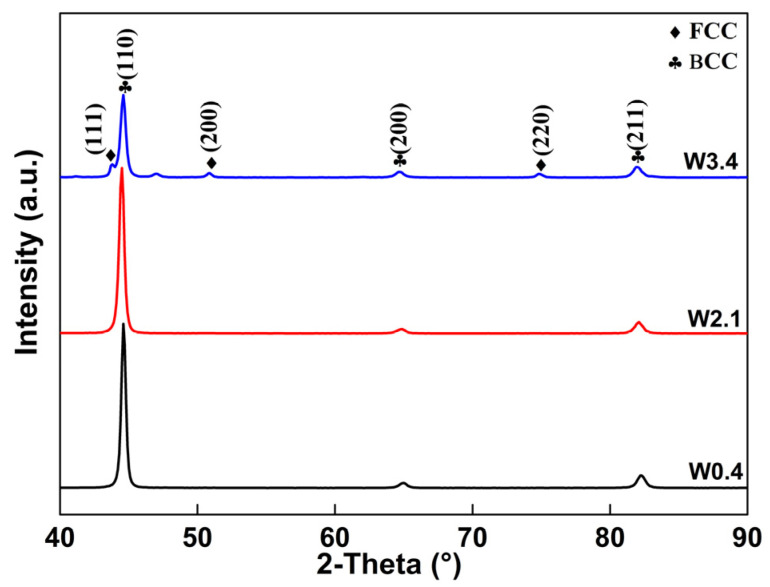
XRD plot of the FeCrCoW alloys with different contents of W.

**Figure 3 materials-16-04319-f003:**
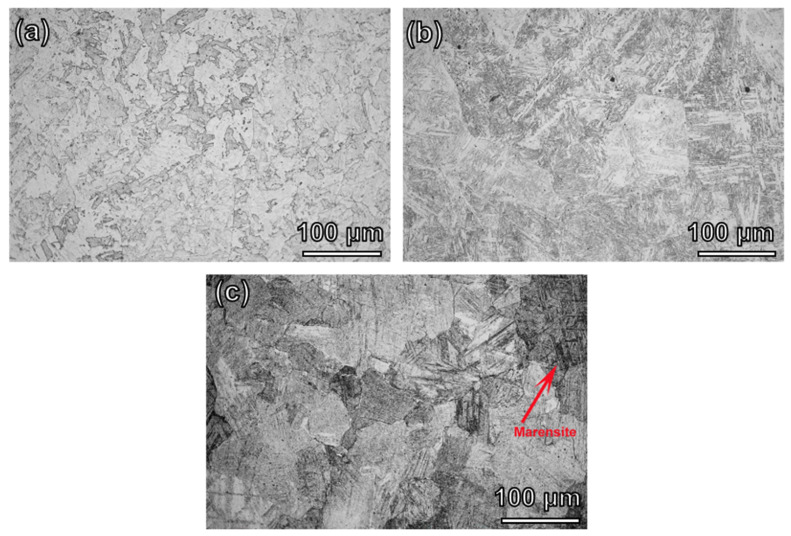
Optical microstructures of the FeCrCoW alloys: (**a**) W0.4, (**b**) W2.1, (**c**) W3.4.

**Figure 4 materials-16-04319-f004:**
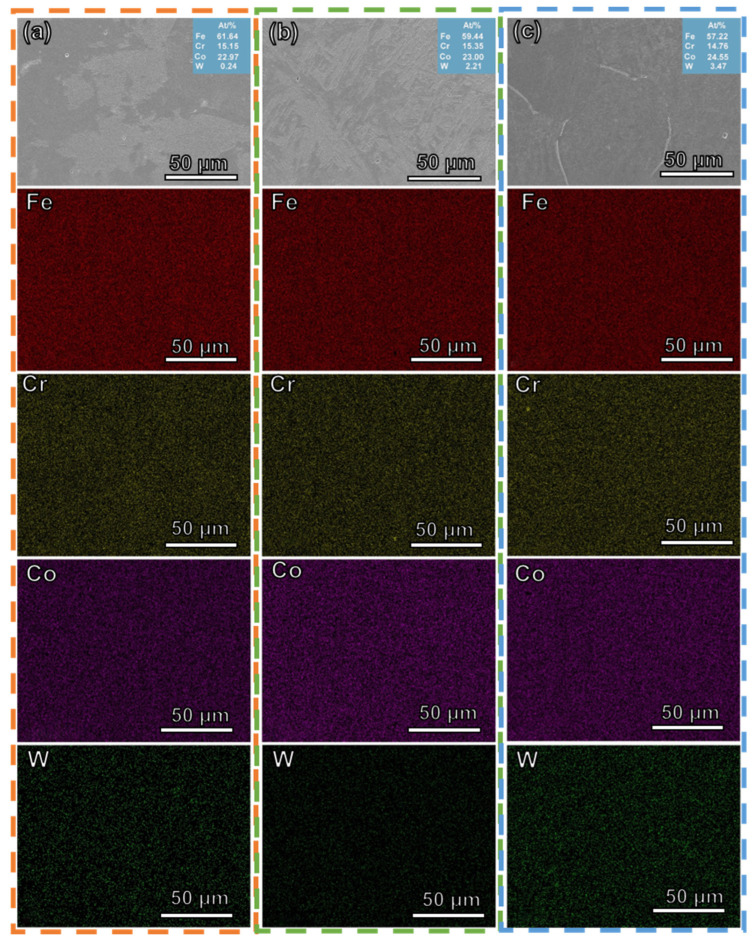
SEM images and EDS mapping for the distribution of all elements (Fe, Cr, Co and W) with different contents in the FeCrCoW alloys: (**a**) W0.4, (**b**) W2.1, (**c**) W3.4.

**Figure 5 materials-16-04319-f005:**
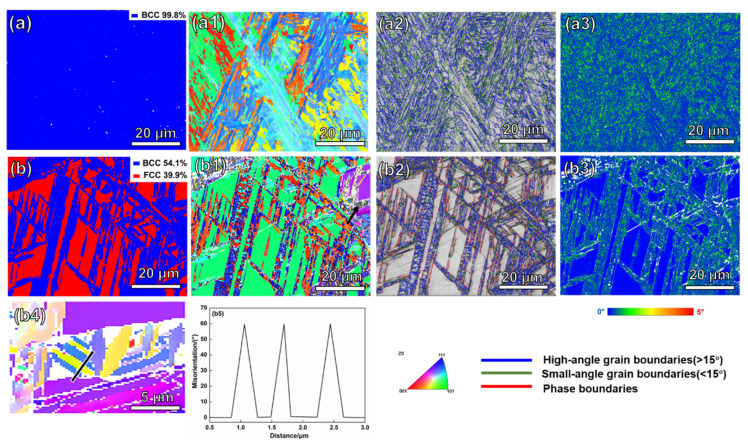
(**a**–**a3**) EBSD maps of the W2.1 alloy and (**b**–**b5**) EBSD maps of the W3.4 alloy, respectively. (**a**,**b**) Phase maps, (**a1**,**b1**) inverse pole figure (IPF) maps, (**a2**,**b2**) grain boundary and phase boundary maps (the blue line represents high-angle grain boundaries, the green line represents small-angle grain boundaries, and the red line represents phase boundaries), and (**a3**,**b3**) KAM diagrams. (**b4**) Orientation imaging image and (**b5**) twin boundary orientation difference image (**b4**).

**Figure 6 materials-16-04319-f006:**
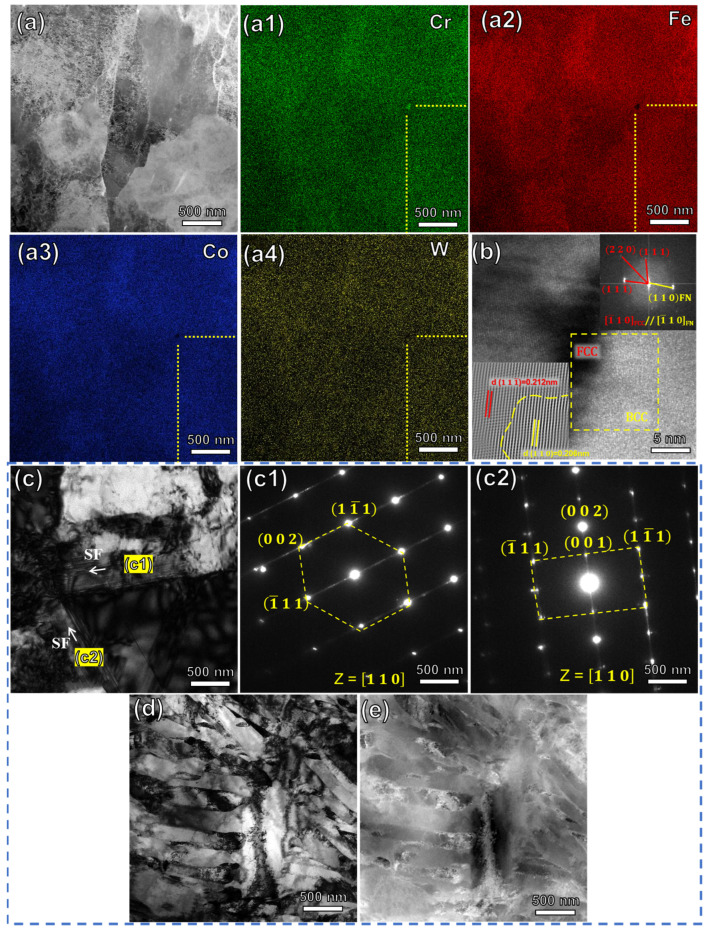
(**a**,**b**) TEM images of the W2.1 alloy sample, (**a**) high-angle annular dark-field (HAADF) scanning, (**a1**–**a4**) EDS mapping images of the enriched Cr particle in the W2.1 alloy, and (**b**) high-resolution TEM (HRTEM) images revealing the phase interface between the enriched Cr particle and the matrix, as well as the corresponding FFT pattern for the interface area in the upper right corner. (**c**–**e**) TEM images of the W3.4 alloy sample, (**c**) cross-stacking faults, (**c1**,**c2**) selected area electron diffraction (SAED) images, and (**d**,**e**) DF and HAADF images of martensite, respectively.

**Figure 7 materials-16-04319-f007:**
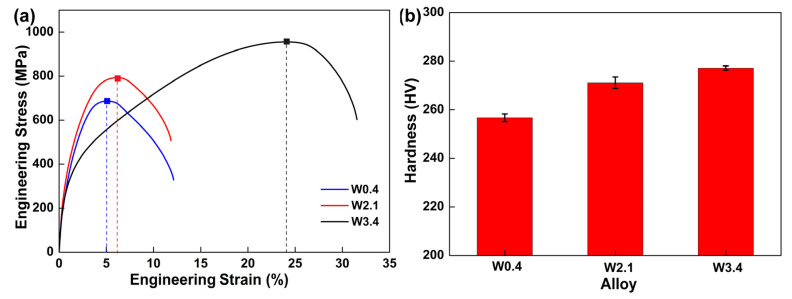
(**a**) Tensile curves of the FeCrCoW alloys at room temperature, (**b**) changing trend of Vickers hardness.

**Figure 8 materials-16-04319-f008:**
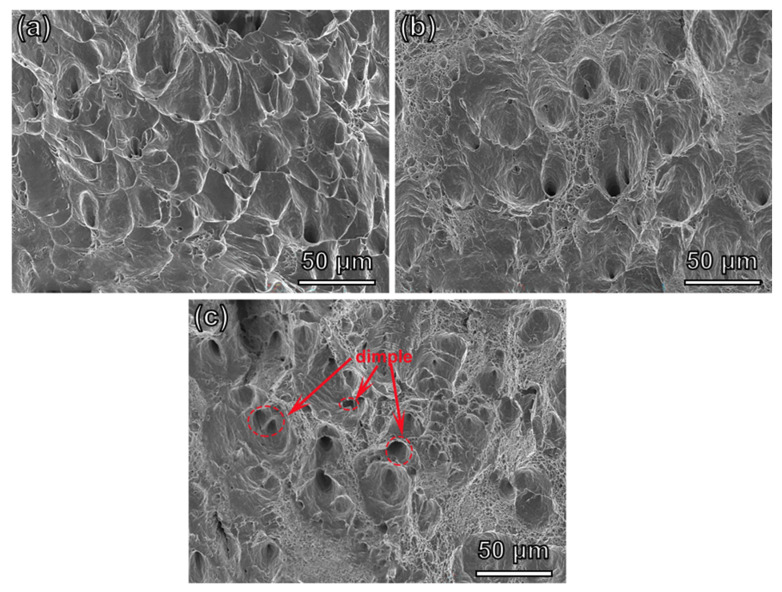
Fracture morphologies of the FeCrCoW alloys: (**a**) W0.4, (**b**) W2.1, (**c**) W3.4.

**Figure 9 materials-16-04319-f009:**
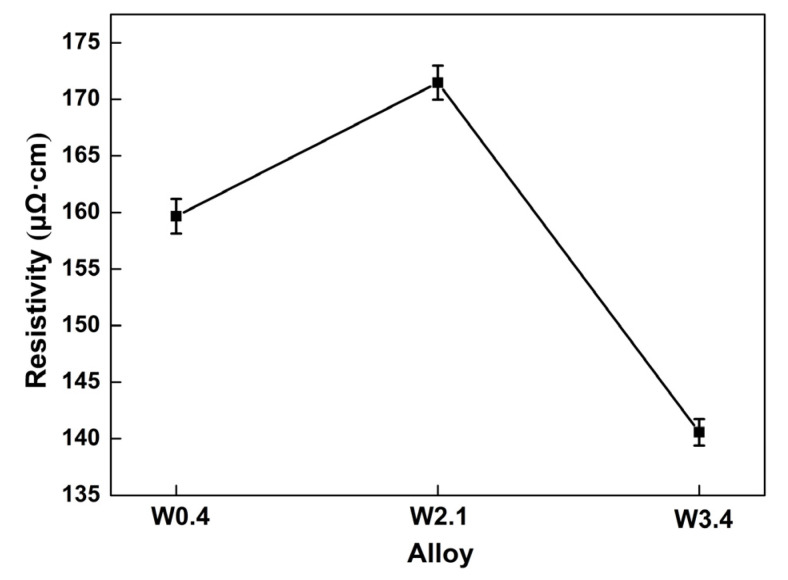
Resistivity with different contents of W in FeCrCo alloys.

**Figure 10 materials-16-04319-f010:**
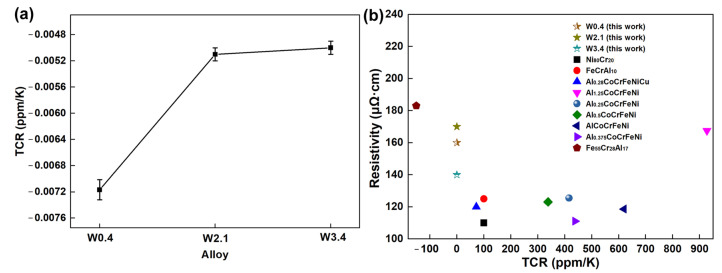
(**a**) TCR of FeCrCo alloys with different W content. (**b**) TCR and resistivity of the FeCrCoxW alloys, compared with commercial alloys such as Ni_80_Cr_20_ [48] and FeCrAl_10_ [48]; some HEAs, such as Al_0.28_CoCrFeNiCu [13], AlxCoCrFeNi [49] and Fe_55_Cr_28_Al_17_ [18], are also shown for comparison.

**Table 1 materials-16-04319-t001:** The chemical composition of the FeCrCoW alloy.

Alloy	W/at%	Cr/at%	Co/at%	Fe/at%
W0.4	0.4	15.3	23.1	Bal.
W2.1	2.1	15.8	23.3	Bal.
W3.4	3.4	16.3	24.0	Bal.

**Table 2 materials-16-04319-t002:** Lattice parameter of each phase in the FeCrCoW alloys.

Alloy	Phase	{hkl}	Lattice Parameter/nm
W0.4	BCC-*α*	(110)	0.2870
W2.1			0.2878
			0.2877
W3.4	FCC-*γ*	(220)	0.3582

**Table 3 materials-16-04319-t003:** Mechanical properties of the FeCrCoW alloys.

Alloy	UTS (MPa)	YS (MPa)	UE (%)
W0.4	687 ± 4.9	237 ± 2.7	5.0 ± 1.2
W2.1	793 ± 3.1	301 ± 3.9	6.3 ± 2.9
W3.4	956 ± 3.3	320 ± 4.1	24.2 ± 2.0

## Data Availability

All data generated or analyzed during this study are included in this published article.

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
