# Peer review of "The Effect of W Content on the Microstructure, Mechanics and Electrical Performance of an FeCrCo Alloy"

_materials, 2023, doi:10.3390/ma16124319_

Round 1
Reviewer 1 Report
A major revision needs for this manuscript based on some issues. Other comments could be found as follows,
1) The novelty of this manuscript is unclear from the title. It is better to mention the effect of the W content.
2) All abbreviations must be defined at first mentioning, especially in the abstract.
3) Testing always needs repeatability. Therefore, all values must have an average data plus the standard deviation, in the whole text.
4) What is “Twin” in keywords, when it could not be found in the abstract or the title? How could it be a keyword? All keywords must be in the abstract or the title.
5) The novelty must be highlighted in the introduction, compared to the literature review.
6) All process parameters need references, such as 1600 C, 900 C, 1 h, etc.
7) What was the standard for testing? Both mechanical and electrical ones? What was the repeatability of testing?
8) All formulations need references.
9) What is the change in XRD results when the W content is changed? What is the effect? What about the microstructure? The quantitative analysis must be done on the microstructure. For example, the grain size, phase content, etc.
10) What is M in Figure 3?
11) From the EDX analysis in Figure 4, what issues could be understood? First of all, the images must be enlarged. Then, no differences could be found for the three different contents of W.
12) The TEM analysis needs more discussion, meaning more descriptions compared to the references.
13) No repeatability of results could be found in Figure 7(a). They must be added to the data. The background color must be white.
14) The standard deviation is not clear in Figure 7(b)! The vertical axis must be from 200 to 300 HV, instead of 0-350 HV.
15) The standard deviation must be added to all data in Table 3. Moreover. What is HV? It must be hardness (HV). In addition, this data could be seen in Figure 7(b). No repeated data could be mentioned in the text. Only one presentation, table or figure.
16) From SEM images in Figure 8, what is the change in the fracture behavior of the material when W content is changed? Moreover, what is the failure mechanism? What is the change in failure mechanism when W changes? The EDX map must be reported to find the failure causes.
17) The standard deviation must be added to Figure 9 with the columnar presentation, such as the hardness. Then, the background color must be white in Figure 9. These issues must be addressed in Figure 10(a).
18) The background color of the figures must be white to find the data clearer.
19) In general, the discussion is not deep. Obtained results must be compared to other results of other articles.
20) No quantitative data could be found in the conclusion.
21) Old references must be removed. Then, they must be updated based on recent articles, published in 2017-2023.
22) What is the difference between this manuscript and the following publications? Do not use them if they are not related, since they are from similar authors and they were not related to the reviewer.
*https://scholar.google.com/scholar?hl=en&as_sdt=2007&q=microstructure%2C+mechanical+and+electrical+performances+of+FeCrCoW+alloy&btnG=
**https://scholar.google.com/scholar?hl=en&as_sdt=2007&q=FeCrCoW+alloy&btnG=
One example: "chemical composition" must be changed to "the chemical composition".
Reviewer 2 Report
In this study ,microstructure, mechanical and electrical properties of FeCrCoW Alloy were discusssed in references to compostional change in W.
Abstract is well written, please specify the application of the research. What is the reason for selection of W as changing compostional element although other constituents are more important. Clarify pls. Does the temperature range is suitable for all industrial applications.
Introdution section is reasonable and relevant. Please add "https://www.frontiersin.org/articles/10.3389/fphy.2019.00097/full " referennce in the introduction section.
Result and discussion : Line 15-158 incomplete sentence. "It is not difficult to found that. W element were dissolved into FeCrCo alloy.
How grain boundry stengthening is effected by concentration of W?
Minor editing and proofread required
Reviewer 3 Report
The manuscript discusses the effect of various W additions on the microstructural, mechanical, and electrical properties of FeCrCo alloy. The manuscript seems interesting and can be accepted after addressing the following comments:
1) Add some quantitative data in the abstract to increase the interest of the reader.
2) What is the motivation behind this study?
3) The quality of Fig.1 and Fig.2 is poor.
4) Why the addition of W result in grain refinement. Please explain more and add some suitable references as well.
5) Add the scale bar in Fig.5.
6) Authors mentioned that
"The BCC phase is blue with a 177 low KAM value, while the FCC phase is green with a high KAM value, as shown in Figure 5 (b3)."
However, it is not correct and the opposite trend can be seen.
7) Please mention the average KAM value for W2.1 and W3.4.
8) "This 202 slip can lead to the occurrence of some dislocations and stacking faults, which can further evolve into twin boundaries."
Please add the suitable reference
Quality of English is satisfactory
Round 2
Reviewer 1 Report
The following comments were answered but not addressed in the revised text, and in some other cases, no proper answers were mentioned, as follows,
Question 6: All process parameters need references, such as 1600℃, 900℃, 1h, etc.
Question 7: What was the standard for testing? Both mechanical and electrical ones? What was the repeatability of testing?
Question 9: What is the change in XRD results when the W content is changed? What is the effect? What about the microstructure? The quantitative analysis must be done on the microstructure. For example, the grain size, phase content, etc.
Question 10: What is M in Figure 3? (It should be mentioned in the figure title!)
Question 12: The TEM analysis needs more discussion, meaning more descriptions compared to the references. (No new references could be found for this part!)
Question 16: From SEM images in Figure 8, what is the change in the fracture behavior of the material when W content is changed? Moreover, what is the failure mechanism? What is the change in failure mechanism when W changes? The EDX map must be reported to find the failure causes.
Question 19: In general, the discussion is not deep. Obtained results must be compared to other results of other articles. (Only adding a figure is not enough. More details of results must be compared to other articles, in the text.)
Round 3
Reviewer 1 Report
Almost done!
Author Response
Dear Editor and Reviewer,
Thank you for your useful comments and suggestions once again on our manuscript entitled “The study on the microstructure, mechanical and electrical performances of FeCrCoW alloy” (Manuscript ID: materials-2382444). Those suggestions are all valuable and very helpful for revising and improving our paper. We have read through the comments carefully and have made corrections. We uploaded the file of the revised manuscript.
We would love to thank you for allowing us to resubmit a revised copy of the manuscript and we highly appreciate your time and consideration. The manuscript has been resubmitted to the journal. We look forward to your positive response.
Best Wishes
Sincerely,
Hui Zhang